# Exploring epidemic control policies using nonlinear programming and mathematical models

Sandra Montes-Olivas[1]*, Adam J. Kucharski[1], Michael B. Gravenor[2], Simon D.W. Frost[1,3]

**1** Department of Infectious Disease Epidemiology and Dynamics, London School of Hygiene & Tropical Medicine, London, United Kingdom, **2** Department of Health Data Science, Faculty of Medicine, Health & Life Science, Swansea University, Swansea, United Kingdom, **3** Microsoft Discovery and Quantum, Microsoft, Redmond, Washington D. C., United States of America

* sandra.montes-olivas@lshtm.ac.uk

## Abstract

Optimal control theory in epidemiology has been used to establish the most effective intervention strategies for managing and mitigating the spread of infectious diseases while considering constraints and costs. Using Pontryagin's Maximum Principle, indirect methods provide necessary optimality conditions by transforming the control problem into a two-point boundary value problem. However, these approaches are often sensitive to initial guesses and can be computationally challenging, especially when dealing with complex constraints. In contrast, direct methods, which discretise the optimal control problem into a nonlinear programming (NLP) formulation, hold potential for automation and could offer suitable, adaptable solutions for real-time decision-making. However, despite their potential, the widespread adoption of these techniques has been limited. Several factors may contribute to this challenge, including limited access to specialised software, a perception of high computational costs, or a general unfamiliarity with these methods. This study investigates the feasibility, robustness, and potential of direct optimal control methods using nonlinear programming solvers on compartmental models described by ordinary differential equations to determine the best application of various interventions, including non-pharmaceutical interventions (NPIs) and vaccination strategies. Through case studies, we demonstrate the use of NLP solvers to determine the optimal application of interventions based on single objectives, such as minimising total infections, "flattening the curve", or reducing peak infection levels, as well as multi-objective optimisation to achieve the best combination of interventions. While indirect methods provide useful theoretical insights, direct approaches may be a better fit for the fast-evolving challenges of real-world epidemiology. By integrating newly available data more quickly, direct methods can enhance the ability to make informed and timely decisions for managing outbreaks effectively.

**Data availability statement:** All code illustrating the optimisation scenarios described in this study is publicly available at http://github.com/EpiRecipes/EpiPolicies.

**Funding:** This work was supported by the Medical Research Council (MRC) under the grant "Building an epidemiological modelling toolkit for epidemic preparedness" (Grant Ref: MR/Z503939/1 to SMO, MBG, SDWF). The funders had no role in study design, data collection and analysis, decision to publish, or preparation of the manuscript.

**Competing interests:** The authors have declared that no competing interests exist.

## Author summary

This study demonstrates the practical advantages of direct optimisation methods in epidemiological modelling when there is a need to identify effective strategies for disease control while balancing constraints. Through case studies, it examines the effort required to adapt compartmental models for optimisation, the time needed to obtain an optimal solution, and the performance of both open-source and licensed tools. The study begins by contrasting indirect and direct methods using a simple infection model. It then illustrates the application of an accessible mathematical programming framework, JuMP, to optimise control strategies aimed at reducing infections, minimising intervention costs under constraints, and managing multiple interventions. Finally, the study compares the efficiency of different optimisation algorithms. The results show that direct methods, aided by readily available tools like JuMP and IPOPT, enable efficient, flexible, and interpretable modelling with minimal additional implementation effort. This work demonstrates how these techniques can support informed and timely decision-making in the early stages of an epidemic.

## Introduction

The study of infectious disease dynamics is crucial for the development and assessment of effective control strategies. Mathematical modelling has become an indispensable tool in this field, providing insights into transmission mechanisms and the potential impact of various interventions. Optimal control theory in epidemiology has been used to establish the most effective intervention strategies for managing and mitigating the spread of infectious diseases while considering constraints and costs [1]. Optimal control can provide a framework to balance common control strategies, such as vaccination and isolation, according to an objective function which can be aimed at minimising infections or intervention costs, enabling decision-makers to weigh the benefits of disease control against potential economic and societal impacts.

The literature on infectious disease modelling using optimal control has grown considerably, particularly in response to the recent coronavirus disease 2019 (COVID-19) pandemic [2]. For instance, Godara et al. [3] applied control theory to a Susceptible-Infected-Recovered (SIR) model to minimise the total number of infected individuals while considering the costs of mitigation efforts. Their findings underscored the importance of strict initial interventions, followed by a gradual relaxation as the epidemic progresses and herd immunity increases. Similarly, Britton and Leskela [4] explored optimal non-pharmaceutical interventions (NPIs) within a SIR framework, emphasising that a single, intense lockdown of short duration, implemented at an optimal time, was the most effective strategy for reducing infections while managing cumulative intervention costs. Other studies have highlighted the practical aspects of different control strategies. For example, Miclo et al. [5] focused on suppression policies to prevent healthcare systems from becoming overwhelmed.

They proposed a "filling the box" strategy that involves intensifying suppression measures as infections approach ICU capacity and relaxing them as pressure on healthcare systems decreases. Meanwhile, Asamoah et al. [6] developed an optimal control model and performed a cost-effectiveness analysis of different combinations of various NPIs, including social distancing, personal hygiene, and disinfection of public spaces.

Beyond single-strain models, Arruda et al. [7] addressed reinfection and multiple viral strains using a multi-strain SEIR model, with a case study of the COVID-19 outbreak. Their study highlighted the need to consider the population's waning immunity and the emergence of new variants when implementing control strategies. Xia et al. [8] proposed a geographically tailored optimal control strategy that underscored the importance of spatial control measures, such as border closures, to contain infectious disease outbreaks effectively. Finally, integrating medical and non-medical interventions, Smirnova [9] suggested the need for control strategies that involve behavioural changes during the early stages of an outbreak and the importance of developing improved control tools, such as vaccines and therapeutics, to apply them as the efficacy of early interventions diminishes over time.

Beyond the recent focus prompted by COVID-19, optimal control theory has been broadly applied in the analysis and management of diverse infectious diseases. For vector-borne diseases, optimal control has been applied to malaria [10,11], dengue fever [12], and other arboviruses like Zika and chikungunya, where strategies extend to innovative biological controls such as the Sterile Insect Technique (SIT) and Wolbachia-infected releases [13]. For example, Asamoah et al. [12], applied optimal control to a dengue fever transmission model incorporating asymptomatic carriers and partial immunity to evaluate the cost-effectiveness of treated bednets, vaccination, treatment, and insecticides. Similarly, optimal control approaches have been explored for rabies management. For instance, Charles et al., [14] integrated optimal control theory into a rabies transmission model involving humans, domestic and free-ranging dogs to evaluate the optimal application of control strategies such as domestic dog vaccination, health promotion and surveillance, public awareness, and post-exposure prophylaxis (PEP). For foot-and-mouth disease (FMD), studies such as that by Wang et al., [15] demonstrate how optimal control can be used to assess control interventions within a bi-seasonal model to minimise infections and costs, with their findings suggesting that reducing transmission among children and isolating older infected individuals are critical for limiting the epidemic.

Together, these studies showcase the versatility and impact of optimal control theory in guiding infectious disease management. They also point out the ongoing challenges in effectively applying these methods, particularly the need for approaches that are both adaptable to real-world complexities and practical for rapid implementation. Addressing these challenges requires a closer look at the methodologies used in optimal control.

Optimal control methods can be classified into two categories: indirect and direct [16]. Indirect methods rely on deriving first-order necessary conditions for optimality, typically using the Pontryagin's Maximum Principle (PMP). The PMP states that if the control $u^*(t)$ and state $x^*(t)$ trajectories are optimal, then there exists a piecewise differentiable adjoint variable $\lambda(t)$ that satisfies conditions defined by the Hamiltonian of the problem [17]. Indirect methods are known for their high numerical accuracy, but have significant challenges. Their reliance on precise initialisation makes convergence highly sensitive to the initial guess, and they often require detailed structural knowledge of the optimal solution, such as *a priori* identifying bang-bang or singular arcs. In bang-bang controls, the optimal strategy switches instantaneously between its minimum and maximum allowable values. This results in a control function that is piecewise constant and characterised by sharp transitions, and potentially difficult to identify. The challenge from a singular arc occurs when the control does not operate at either extreme. Instead, it is determined by higher-order conditions, which require solving additional equations to characterise the control trajectory within that specific interval. These complexities introduce extra analytical and computational challenges. As a result, indirect methods can become intricate and time-consuming, particularly for complex or poorly understood models where the solution structure is not immediately apparent.

In contrast, direct methods adopt a more intuitive approach by discretising the model in time, transforming the continuous optimal control problem into a nonlinear optimisation problem (NLP) [18]. A NLP is an optimisation problem in which at least

one component of the formulation, that is, the objective function or the constraints, is a nonlinear function of the decision variables. Formally, an NLP seeks to minimise or maximise a scalar objective function subject to equality and/or inequality constraints, where these functions are not restricted to being linear. The decision variables include the state variables (e.g., $S$, $I$, $R$) and control variables (e.g., intervention intensity) at each discretised time point. The NLP solver then determines values for these variables that minimise (or maximise) the objective while satisfying all constraints. If the discretised NLP is differentiable, it can be addressed using gradient-based optimisation solvers such as the Interior Point Optimizer (IPOPT) solver [19] or sequential quadratic programming (SQP) solvers [20]. Non-differentiable problems can alternatively be solved by non-gradient-based methods, including metaheuristic algorithms such as Genetic Algorithms [21] and Simulated Annealing [22,23].

Recent advances have expanded the optimisation landscape for mechanistic epidemiological models through the integration of deep learning techniques. Colas et al. [24] proposed a toolbox in Python aimed at bridging epidemiological models and reinforcement learning, specifically leveraging Q-Learning combined with deep neural networks (DQN) [25] alongside evolutionary optimisation methods such as NSGA-II [26]. Furthermore, Yin et al. [27] proposed a hybrid optimisation approach integrating traditional optimisation frameworks with deep-learning algorithms, employing both first- and second-order optimisers such as Adam [28,29] and L-BFGS [20,30]. Although these methods show potential, their computational demands can create challenges and may limit their applicability in time-critical scenarios.

In contrast, established NLP methods are often characterised by rapid convergence to locally optimal solutions thus minimising computational demands. These methods can handle complex system dynamics, multiple constraints, and continuously evolving models, making them well suited for real-time intervention planning and decision-making. However, despite their practical advantages and the fact that such methods and solvers are well established within the optimisation community [18,19,31], they have yet to be widely adopted in the context of epidemiological modelling. Factors such as limited access to specialised software, perceived high computational costs, and unfamiliarity with these techniques may hinder their widespread use. Additionally, practical challenges, like adapting complex epidemiological models to fit into an optimal control framework, can create barriers to broader adoption.

This study aims to demonstrate the practical value and accessibility of direct optimal control methods in epidemiological modelling. While the underlying optimisation techniques are well established, their application in epidemiology has been limited by practical challenges such as computational complexity, model adaptability, and accessibility of suitable tools. Through case studies, we demonstrate that direct optimisation methods provide a practical and accessible alternative to indirect approaches, such as forward–backward sweep algorithms, for computing optimal control strategies in epidemiological models. In particular, we show that widely used compartmental models from the literature can be readily adapted to a direct optimisation framework with minimal structural modification. Initial explorations have been conducted to assess the resources required for implementing these methods and to evaluate their reliability. By highlighting both the strengths and limitations of these approaches, this work seeks to lower the barrier to adoption and encourage their use in real-time intervention planning and decision-making for infectious disease management.

## Materials and methods

### Optimisation algorithms

Epidemiological control problems frequently present nonlinear dynamics, and are constrained by the available resources [32]. Based on the existing literature, many epidemiological models that employ optimal control theory rely on the PMP to derive necessary conditions, which are then solved using numerical methods. Among these, the forward-backward sweep method [17] is one of the most commonly employed approaches for simulations. In contrast, direct optimisation methods are widely used in several engineering disciplines to find optimal solutions to complex problems [18]. In these fields, a direct optimisation approach formulates the entire optimisation problem (states, controls, and constraints) in a single framework, as a NLP for instance, which is then solved numerically. The NLP solver methods can be grouped as sequential or simultaneous. Sequential methods, like SQP, involve breaking down the problem into a sequence of subproblems that are updated from previous

iterations. On the other hand, simultaneous methods solve the optimisation problem as a whole by discretising both the state and control profiles in time, which may be better suited for large scale problems [33]. IPOPT is a commonly used open-source solver that is based on an interior-point approach to solve large-scale problems by iteratively improving a candidate solution from within the feasible region [19]. Here, IPOPT was chosen for its proven ability to efficiently solve large-scale nonlinear programming problems, and its wide availability across multiple programming environments.

JuMP (Julia Mathematical Programming) is an open-source algebraic modelling language embedded in Julia [31,34]. Similar to other commercially available optimisation software such as AMPL (A Mathematical Programming Language) [35] and GAMS (General Algebraic Modeling System) [36], JuMP enables users to express optimisation problems in a form similar to their original mathematical representation, which is then translated into the form expected by the solver. JuMP provides an interface to various solvers and supports automatic differentiation, making it highly efficient for large-scale optimisation problems. The main components of a NLP formulation in JuMP are as follows; a solver; a list of variables; constraints that define the feasible region for the solution; and the objective function used to minimise or maximise specific outcomes, such as the number of infections or costs.

All models presented in this study were discretised in time using a basic Euler method, a straightforward and widely used approach for approximating solutions. This discretisation helps maintain consistency in the behaviour of the model and transforms the continuous system into a form suitable for solving the NLP. The IPOPT solver was used to solve the NLP due to its open-source nature and broad support across the optimisation languages used in this study. The resulting NLPs were solved using `IPOPT` (v3.14.17), interfaced through `JuMP` (v1.24), chosen for its open-source availability and broad support across optimisation frameworks. Default solver settings were employed, including a convergence tolerance of $10^{-6}$, a feasibility tolerance of $10^{-6}$, and a maximum of 3000 iterations. Linear systems were solved using MUMPS, and IPOPT's default exact Hessian of the Lagrangian was used. All case studies converged successfully, with constraint violations remaining below the specified feasibility tolerance, and no manual scaling or problem reformulation was required.

## Model formulation

To demonstrate the application of direct optimisation methods, we begin with a simple exponential-growth infection model in which a time-dependent control intervention seeks to reduce the number of infected individuals. This example provides a setting to compare the two methodologies: the indirect method, in which the Pontryagin's Maximum Principle is applied to derive optimality conditions, and the direct method, in which the problem is discretised and solved as a nonlinear programming problem.

Next, we present the application of JuMP with IPOPT across four case scenarios that employ models already established within epidemiological research [4,12]. The first three scenarios evaluate single control intervention strategies on a modified SIR model aimed at minimising total infections in an epidemic through lockdown measures, 'flattening the curve,' or vaccination efforts. The final scenario evaluates a more complex compartmental model of dengue transmission formulated by Asamoah et al. [12], optimising the combination of four control strategies.

All case scenarios were solved on a MacBook Pro with an Apple M3 Pro chip and 18GB of memory. The computational framework employed JuMP (v1.24), IPOPT (v3.14.17), AMPL (v20241203), amplpy(v0.14.0), rAMPL (v2.0.13.0.20241012), AmplNLWriter (v1.2.3) and Pyomo (v6.8.2). Code illustrating these scenarios can be found at http://github.com/EpiRecipes/EpiPolicies.

## Results

### Comparison of indirect and direct methodologies

We consider a simple mechanistic model that tracks the number of infected individuals $I(t)$ over time, assuming a nearly constant susceptible population $S \approx N$, where $N$ denotes the total population size. The dynamics of infection are described by:

$$\frac{dI}{dt} = \left( \beta(1 - \upsilon(t))N - \gamma \right) I,$$

(1)

where $\beta$ is the transmission rate, $\gamma$ is the recovery rate, and $v(t) \in [0, v_{max}]$ represents a time-dependent control strategy (i.e., varying social distancing measures or lockdown) aimed at reducing transmission.

The optimal control policy $v(t)$ that balances disease mitigation with the cost of intervention is defined by the following objective function:

$$J = \int_0^{T_f} \left[ AI(t) + B v(t)^2 \right] dt, \tag{2}$$

where $A$ and $B$ are the infection and control effort weights, respectively.

First, PMP was used to construct the Hamiltonian:

$$\mathcal{H}(I, v, \lambda) = AI + Bv^2 + \lambda \left[ (\beta(1-v)N - \gamma)I \right], \tag{3}$$

where $\lambda(t)$ is the adjoint variable associated with state $I(t)$. Then, the adjoint equation is given by:

$$\frac{d\lambda}{dt} = -\frac{\partial\mathcal{H}}{\partial I} = -A - \lambda \left( \beta(1-v)N - \gamma \right), \tag{4}$$

To obtain the optimal control, we solve the following condition:

$$\frac{\partial\mathcal{H}}{\partial v} = 2Bv - \lambda\beta NI = 0, \tag{5}$$

which yields the unconstrained optimal control:

$$v^*(t) = \frac{\lambda(t)\beta NI(t)}{2B}. \tag{6}$$

Taking into account control bounds, the final expression for the optimal control is as follows:

$$v^*(t) = \min \left( \max \left( 0, \frac{\lambda(t)\beta NI(t)}{2B} \right), v_{max} \right). \tag{7}$$

Using these derivations, the optimal control problem was solved numerically using the forward-backward sweep method [17], which iteratively updates the state and adjoint equations until convergence.

To compare with a direct approach, the optimisation problem was then formulated and solved using the algebraic modelling language JuMP [34] with the the interior-point solver IPOPT [19]. This approach requires the user to specify a solver and define decision variables, constraints, and the objective function. JuMP then translates the algebraic representation of the problem into a standard form that the solver interprets to find an optimal solution.

Code snippets illustrating the implementation of both methods are shown in Fig 1. The direct method not only results in a more concise block of code, but also eliminates the need for extensive preparatory analytical work, which can be considerable when applied to more complex models, compared to the simplified example presented here. Crucially, Fig 2 presents the results obtained from each approach, showing that both methods produced the same optimal solution.

**Case scenario 1: Lockdown.** This scenario examines the optimal control of an SIR model through a lockdown intervention that reduces the infection rate, with the aim of minimising the total number of infected individuals. The state variables are as follows: $S$ represents the number of susceptible individuals, $I$ denotes the number of infected individuals,

```
1     while test < 0 && sweep < max_iter
2       sweep += 1
3
4       I_old = copy(I)
5       v_old = copy(v)
6       λI_old = copy(λI)
7
8       for k in 1:T
9        infection = dt * β * (1 - v[k]) * N
              * I[k]
10       recovery  = dt * γ * I[k]
11       I[k+1] = I[k] + infection - recovery
12      end
13
14      λI[T+1] = 0.0
15      for k in T:-1:1
16        λI[k] = λI[k+1] + dt * (-A + λI[k+1]
              * (β * (1 - v[k]) * N - γ))
17      end
18
19      temp = -λI.*β.*N.*I./(2 .* B)
20      v_new = clamp.(temp, 0.0, v_max)
21      v .= 0.5 .* (v_new .+ v_old)
22
23      test = minimum([
24        δ * sum(abs.(v)) - sum(abs.(v .- v
              _old)),
25        δ * sum(abs.(I)) - sum(abs.(I .-
              I_old)),
26        δ * sum(abs.(λI)) - sum(abs.(λI .- λ
              I_old))])
27      end
```

**1.** Forward-backward sweep

```
1  using JuMP, Ipopt
2  model = Model(Ipopt.Optimizer)
3  set_optimizer_attribute(model, "max_iter",
       1000)
4  @variable(model, 0 <= I[1:(T+1)] <= 1)
5  @variable(model, 0 <= v[1:(T+1)] <= v_max)
6  @expressions(model, begin
7   infection[t in 1:T], (1 - v[t]) * β * N * I[t
       ] * dt
8   recovery[t in 1:T], γ * dt * I[t]
9      end)
10 @constraints(model, begin
11  I[1]==I0
12  [t=1:T], I[t+1] == I[t] + infection[t] -
       recovery[t]
13     end)
14 @objective(model, Min,
            sum(dt * (A * I[t] + B * v[t]^2)
               for t in 1:T+1))
16 optimize!(model)
```

**2.** JuMP + IPOPT

**Fig 1. Code Snippets: FBS vs JuMP–Ipopt.** Code snippets implementing the different approaches to solve the optimal control problem: forward-backward sweep (left) and JuMP with IPOPT solver (right).

and $C$ indicates the total number of cases. The infection rate is modified according to a policy denoted as $v(t)$, where $0 \leq v(t) \leq v_{max} \leq 1$. This model is described by the following differential equations (Eq (8)):

$$\frac{dS}{dt} = -\beta(1 - v(t))SI,$$
$$\frac{dI}{dt} = \beta(1 - v(t))SI - \gamma I,$$
$$\frac{dC}{dt} = \beta(1 - v(t))SI$$

(8)

where $\beta$ and $\gamma$ are the baseline transmission and recovery rates, respectively.

Britton and Leskela [4] analytically demonstrated that the optimal policy under these conditions is a single "bang-bang" intervention, characterised by a sudden shift from no intervention to the maximum allowable level for a single period. The optimal control problem is defined as the policy that minimises the total number of cases (i.e., $C(\infty)$) while adhering to two main constraints: (a) the value of $v$ cannot exceed a specified maximum, $v_{max}$, and (b) there is a cost associated with the policy, quantified as the integral of $v$ over time, which must remain within a certain limit, i.e., $\int v(t)dt \leq v_{total}$. In the implementation, we calculate $C$ for a long time horizon to approximate $C(\infty)$.

The parameter values used in the evaluations with the modified SIR models are listed in Table 1, and they were assumed for the purpose of intervention optimisation only. These values were not derived from real-world data but were

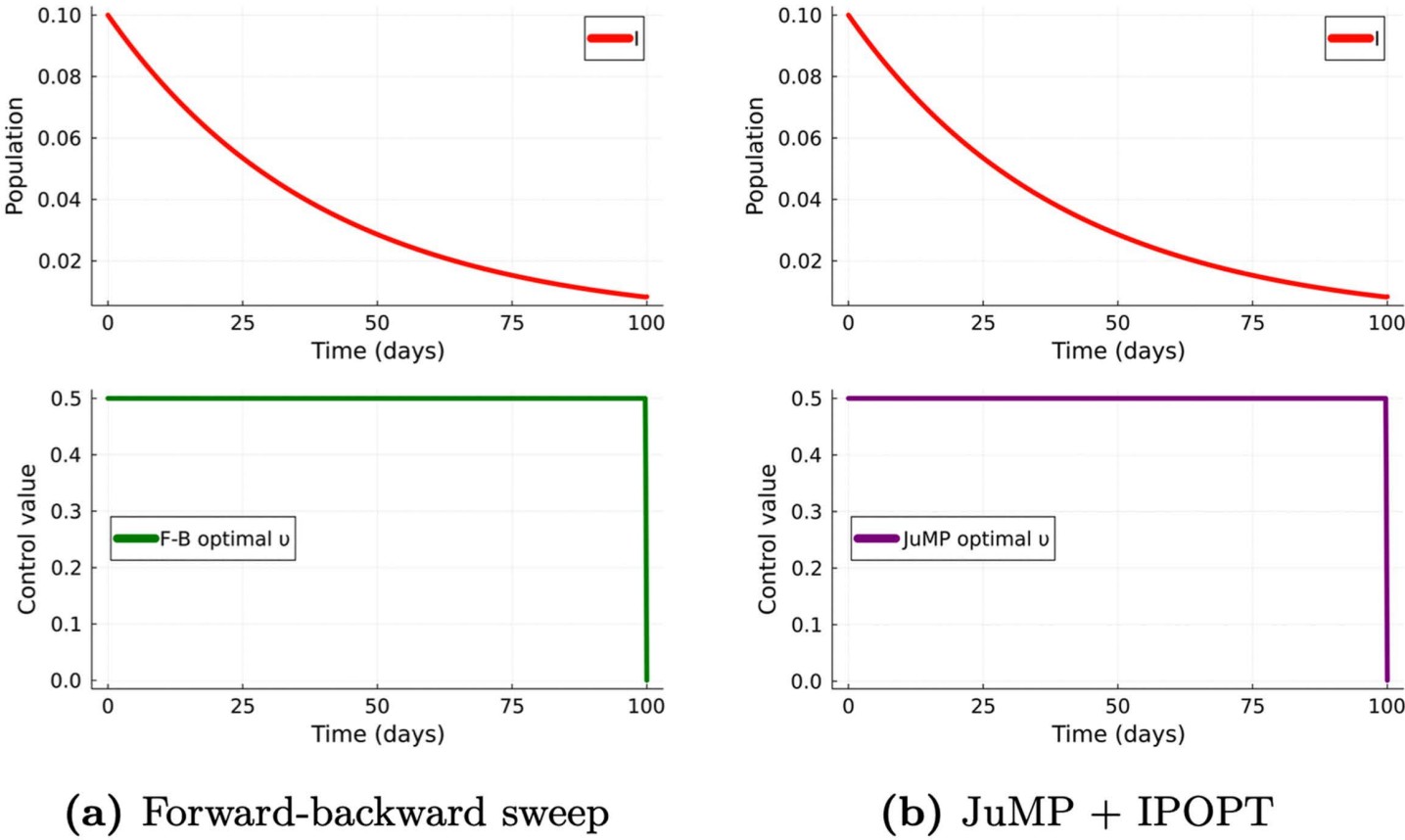

**Fig 2. Optimal control solutions: FBS vs JuMP–Ipopt.** Optimal control solutions obtained using the (a) forward-backward sweep method and the (b) JuMP modeling framework with IPOPT.

instead utilised to assess the performance of the optimisation algorithm within the model. We initially ran the model without the presence of intervention to establish baseline outputs, observing a cumulative incidence of approximately 79% and the peak of infection occurring at around 17.5 days (see Fig 3(a)).

This information was used to simulate the impact of a heuristic (non-optimal) baseline intervention strategy: a single lockdown intervention initiated at the peak of infection cases and lasting for a fixed period of 20 days (set at the maximum policy value of 0.5) consistent with the duration suggested by Britton and Leskela [4]. This scenario was chosen as representative of an intervention with a simple operational definition. Fig 3(b) shows that, in this case, the final cumulative incidence under intervention is 63%.

To investigate whether these results represent the optimal values for minimising cumulative incidence in this scenario, the NLP model was run. The results, shown in Fig 4(a), indicate that the optimal start time obtained using JuMP is 14.3 days, which is earlier than the peak observed in the baseline non-intervention model (17.5 days). Furthermore, the optimal output suggests implementing a single lockdown lasting approximately 19.9 days, and the final cumulative incidence is 59%.

**Case scenario 2: "Flattening the curve".** This scenario employs the same system of equations (Eq (8), as in the lockdown scenario. In addition, in this case, the optimal intervention policy, represented as $v(t)$, balances the costs associated with intervention against the need to manage the spread of infection. This policy aims

**Table 1. Values of parameters used in evaluations performed using the modified SIR models.**

| Parameter | Value | Definition |
|---|---|---|
| $\beta$ | 0.5 | Transmission rate |
| $\gamma$ | 0.25 | Recovery rate |
| S(0) | 0.99 | Initial fraction of suceptible population |
| I(0) | 0.01 | Initial fraction of infected population |
| $v_{max}$ | 0.5 | Maximum policy value |
| $v_{total}$ | 10 | Budget set for the control policy |
| $I_{max}$ | 0.5 | Scenario 2: infected population threshold value |

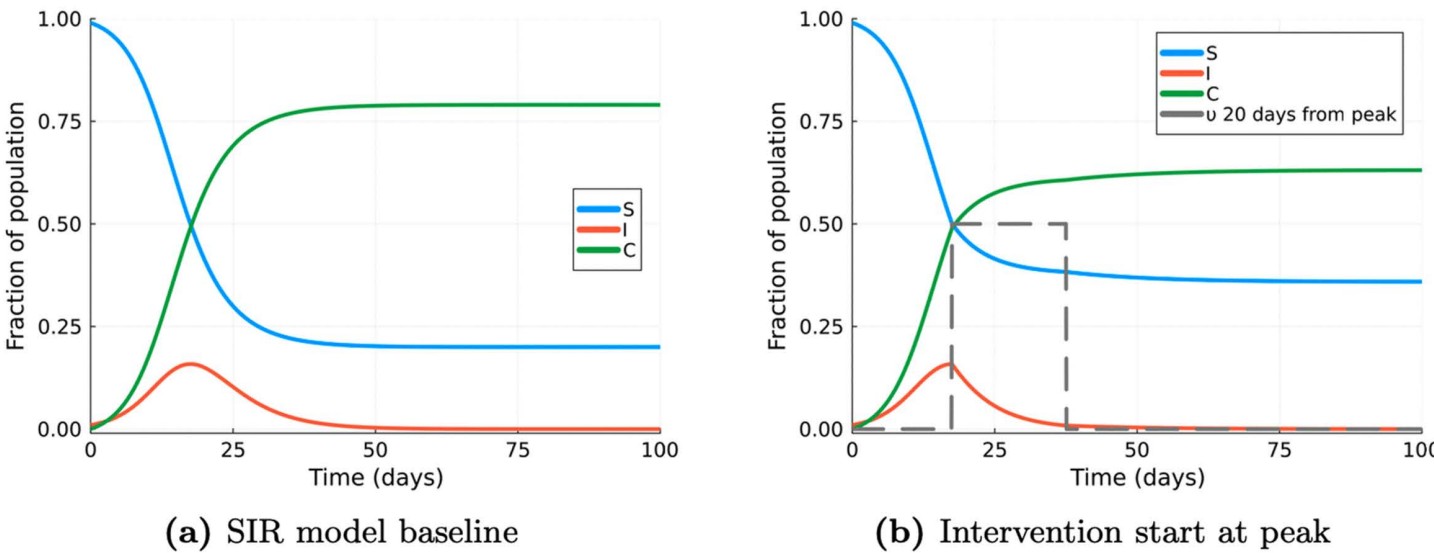

**(a) SIR model baseline**   **(b) Intervention start at peak**

**Fig 3. SIR baseline vs intervention starting at peak of infections (non-optimal).** Comparison of SIR model outputs (a) baseline scenario without intervention and (b) with a lockdown intervention that lasts 20 simulated days, set at $v_{max}$ and applied at the peak of infections.

to ensure that the number of infected individuals, denoted as *I*, does not exceed a predetermined threshold, $I_{max}$. This approach is commonly referred to as "flattening the curve" (FtC), and it is easily implemented in the constraints included in the NLP. Both $v(t)$ and *I(t)* are constrained to values between 0 and $v_{max}$ and $I_{max}$, respectively. The objective function in this scenario is to minimise the total co*st* of intervention instead of the cumulative incidence.

The results shown in Fig 4(c) suggest that the optimal policy following a FtC approach involves a single lockdown that increases in intensity rapidly at or shortly before the maximum tolerable infections ($I_{max}$). Once this threshold is reached the intensity of the lockdown is gradually reduced. Fig 4(d) shows that the end of the lockdown coincides with the effective reproduction number, $R_t$, crossing the value of 1. This ensures that the population of infected individuals remains below the critical level.

**Case scenario 3: Vaccination.** From an epidemiological standpoint, in vaccination-based control strategies, the primary goal is to reduce the susceptible population below a threshold required for sustained transmission. This process is mathematically represented by Eq (9), where the main difference is the position of the control variable $v(t)$, which in this case represents vaccination at a per-capita rate.

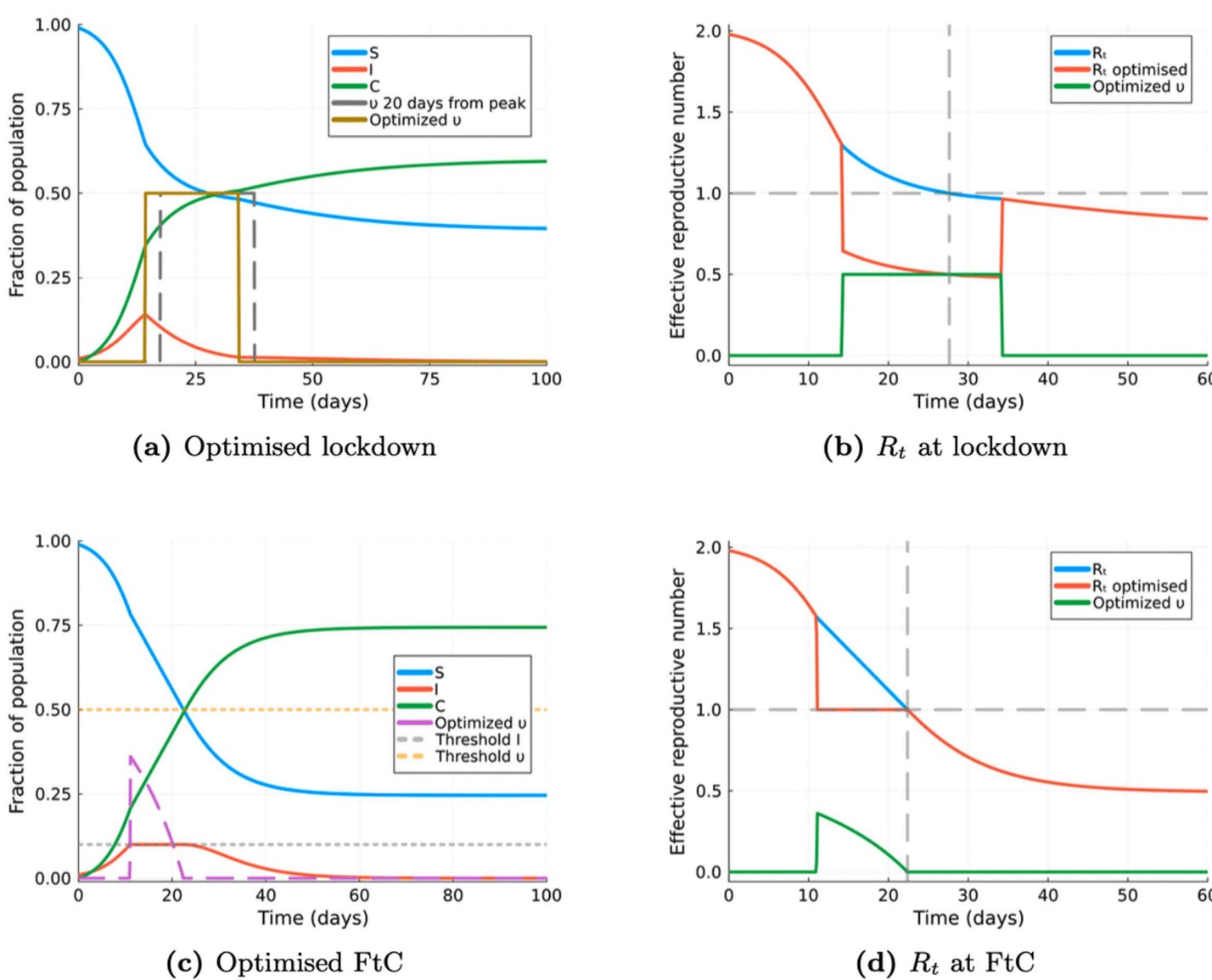

**Fig 4. SIR with optimised NPIs.** Comparison of SIR model outputs (a) optimised lockdown scenario, (b) effective reproduction number during lockdown, (c) optimised "flattening the curve" (FtC) scenario, and (d) effective reproduction number during FtC intervention.

$$\frac{dS}{dt} = -\beta SI - \upsilon(t)S,$$
$$\frac{dI}{dt} = \beta SI - \gamma I,$$
$$\frac{dC}{dt} = \beta SI \tag{9}$$

The optimal control problem is defined as the policy that minimises the total number of cases (i.e., the final size of the epidemic) while adhering to the following constraints: (a) the vaccination rate, $\upsilon$, cannot exceed a maximum value, indicating

a limit on the rate of vaccination, and (b) there is a cost associated with the vaccination process, measured as the integral of $v(t) * S(t)$ over time, which cannot exceed a predetermined level.

The directly optimised results shown in Fig 5 suggest that the optimal policy in this scenario is initiating vaccination immediately, and that maintaining a continuous administration at the maximum allowable rate is most effective. This intervention should be maintained until the allocated vaccine resources are depleted. This strategy reduces the susceptible population early in the epidemic, significantly limiting disease transmission and minimising the peak incidence of infections to ~3%.

**Case scenario 4: Multiple control strategies.** To assess the effectiveness of direct optimisation methods in identifying the optimal solution for a combination of multiple interventions, we applied the same methodology to a model proposed by Asamoah et al. [12], which provides a framework for evaluating the combination of different control interventions to control dengue transmission. This model considers two populations: humans and female mosquitos. The human population is divided into five compartments: susceptible $S_h$, infected (symptomatic) $I_h$, carrier (asymptomatic) $I_{hA}$, partially immune $P$, and recovered $R_h$, while the mosquito population is formed by two compartments: susceptible $S_v$ and infected $I_v$. The model is described by Eq (10, 11).

$$N_h(t) = S_h(t) + I_h(t) + I_{hA}(t) + P(t) + R_h(t),$$
$$N_v(t) = S_v(t) + I_v(t),$$
$$\lambda_h(t) = \frac{(1 - u_1(t))b\beta_1}{N_h(t)} I_v(t),$$
$$\lambda_h 1(t) = \frac{(1 - u_1(t))b\beta_2}{N_h(t)} I_v(t),$$
$$\lambda_v(t) = \frac{b\beta_3}{N_h(t)} (I_h(t) + I_{hA}(t)).$$

(10)

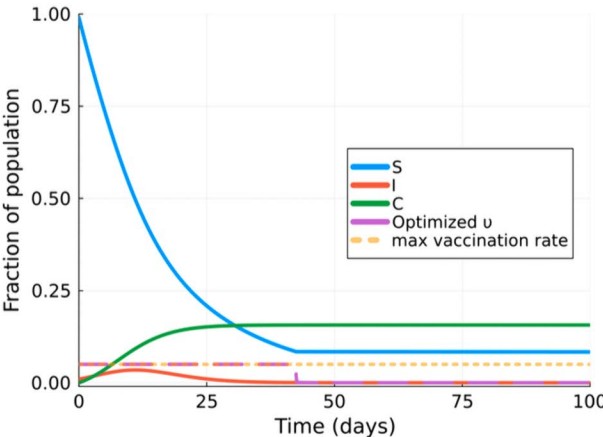

**Fig 5. SIR with optimised vaccination.** Comparison of SIR model outputs (a) baseline scenario without intervention and (b) with a lockdown intervention that lasts 20 simulated days, set at $v_{max}$ and applied at the peak of infections.

$$\frac{dS_h}{dt} = \mu_h N_h - \lambda_h S_h - S_h u_2 - \mu_h S_h,$$

$$\frac{dI_h}{dt} = \psi \lambda_h S_h + \omega \lambda_h 1 P - (\mu_h + u_3 + \gamma_h) I_h,$$

$$\frac{dI_{hA}}{dt} = (1 - \psi) \lambda_h S_h + (1 - \omega) \lambda_h 1 P - (\mu_h + \gamma_h) I_{hA},$$

$$\frac{dP}{dt} = u_2 S_h + \rho u_3 I_h + \phi \gamma_h (I_h + I_{hA}) - \lambda_h 1 P - \mu_h P,$$

$$\frac{dR_h}{dt} = (1 - \rho) u_3 I_h + (1 - \phi) \gamma_h (I_h + I_{hA}) - \mu_h R_h,$$

$$\frac{dS_v}{dt} = \mu_v N_v (1 - u_4) - \lambda_v S_v - \mu_v S_v - r_0 u_4 S_v,$$

$$\frac{dI_v}{dt} = \lambda_v S_v - \mu_v I_v - r_0 u_4 I_v. \tag{11}$$

The time-dependent controls in the model are: treated bednets ($u_1$), vaccination ($u_2$), treatment with prophylactics ($u_3$), and insecticides ($u_4$). All parameter definitions and values were obtained from [12], and restated in Table 2 for reference.

Following the Asamoah et al. [12] study, the objective was to minimise dengue incidence and the intervention cost according to the objective function shown in Eq (12), with weights relevant to diferent compartments ($C_{1\ldots3}$), and those relative to each control variable ($D_{1\ldots4}$).

$$\min \sum_{t=1}^{T} C_1 \cdot I_h[t] + C_2 \cdot I_{hA}[t] + C_3 \cdot \left( S_v[t] + I_v[t] \right)$$

$$+ \frac{1}{2} (D_1 u_1^2 + D_2 u_2^2 + D_3 u_3^2 + D_4 u_4^2) \tag{12}$$

Similar to the case scenarios presented previously, the system of equations was discretised using a simple Euler discretisation and optimised using JuMP and the IPOPT solver. All tested control combinations resulted in an optimal solution, and the number of iterations and the time taken to reach this solution are displayed in Table 3. These results indicate that the JuMP optimisation framework, in conjunction with the IPOPT solver, can return optimal solutions, even in more complex scenarios involving the optimisation of multiple control strategies. In this case, where a weighted sum is used to aggregate the contributions of each strategy to the objective function, the algorithms effectively converge to the optimal solution in less than 30 seconds for all combinations tried (see Table 3).

In the original study by Asamoah et al. [12], the optimal controls were obtained using the indirect method based on PMP, as described previously. For this model, the Hamiltonian takes the form:

$$\mathbf{H} = L + \lambda_{S_h} \frac{dS_h}{dt} + \lambda_{I_h} \frac{dI_h}{dt} + \lambda_{I_{hA}} \frac{dI_{hA}}{dt} + \lambda_P \frac{dP}{dt}$$

$$+ \lambda_{R_h} \frac{dR_h}{dt} + \lambda_{S_v} \frac{dS_v}{dt} + \lambda_{I_v} \frac{dI_v}{dt}, \tag{13}$$

where $L$ is the integrand of the objective function (Eq 12), $\lambda_{S_h}, \lambda_{I_h}, \ldots, \lambda_{I_v}$ are the adjoint variables associated with each state, and the derivatives are given by the state equations (Eq 11).

The necessary optimality conditions yield a system of seven adjoint differential equations, $\dot{\lambda}_i = -\partial \mathcal{H} / \partial x_i$, with transversality conditions $\lambda_i(T) = 0$, together with the characterisation of each optimal control via $\partial \mathcal{H} / \partial u_j = 0$. Given the complexity of the model, the resulting adjoint equations expand into extensive expressions (see pages 10–11 in [12]). This highlights

**Table 2. Parameter values and descriptions used on the dengue fever model.**

| Parameter | Value | Definition |
|---|---|---|
| $\beta_1$ | 0.75 | Transmission probability from $I_v$ to $S_h$ |
| $\beta_2$ | 0.375 | Transmission probability from $I_h$ to $S_v$ |
| $\beta_3$ | 0.75 | Transmission probability from $I_v$ to $P$ |
| $b$ | 0.5 | Avg. biting rate per mosquito per person |
| $\rho$ | 0.01 | Proportion of treated individuals with partial immunity |
| $\psi$ | 0.4 | Proportion of incidence rate from $S_h$ to $I_h$ |
| $\gamma_h$ | 0.3288 | Human's disease related death rate |
| $\omega$ | 0.54 | Proportion of incidence rate from P to $I_h$ |
| $\mu_h$ | 0.0045 | Human's natural mortality and recruitment rate |
| $\mu_v$ | 0.0323 | Vector's natural mortality and recruitment rate |
| $\phi$ | 0.48 | Proportion of natural recovery |
| $r_0$ | 0.005 | Enhanced death rate |
| $u_1$ | {0, 0.75} | Treated bednet control strategy |
| $u_2$ | {0, 0.75} | Vaccination control strategy |
| $u_3$ | {0, 0.75} | Treatment with prophylactics control strategy |
| $u_4$ | {0, 0.75} | Insecticides control strategy |
| $C_{1,2,3}$ | 5 | Weights related to $I_h$, $I_{hA}$, $S_v$ and $I_v$ populations |
| $D_1$ | 16.62 | Weight related to $u_1$ |
| $D_2$ | 2.5 | Weight related to $u_2$ |
| $D_3$ | 5 | Weight related to $u_3$ |
| $D_4$ | 16.62 | Weight related to $u_4$ |

**Table 3. Comparison of the number of iterations and the time taken to reach an optimal solution when combining control strategies using JuMP with the IPOPT solver.**

| Interventions | No. Iterations | Time (s) |
|---|---|---|
| $u_1 + u_4$ | 1201 | 24.252 |
| $u_2 + u_3$ | 534 | 9.843 |
| $u_1 + u_3 + u_4$ | 1432 | 27.629 |
| $u_1 + u_2 + u_4$ | 1229 | 23.682 |
| $u_1 + u_2 + u_3 + u_4$ | 1171 | 22.053 |

the substantial analytical effort required by the indirect method when applied to complex systems. We applied the forward-backward sweep method to this model and confirmed that both this and JuMP approaches produced equivalent optimal control trajectories. The code and trajectory comparisons are available in the associated repository.

**Comparison between optimisation algorithms.** There could be concerns regarding the use of direct optimisation methods if these are limited to specific programming languages. To address this, we leveraged the accessibility of the IPOPT solver across various platforms.

Table 4 shows the number of iterations and time each algorithm took to find an optimal solution for the lockdown model. The results indicate that the solver maintains a consistent number of iterations and the resulting objective value across platforms. The timing results shown in the table reflect solve times after JIT compilation warmup for JuMP, ensuring fair comparison across platforms. Overall solve times were consistent across platforms, though JuMP reported a denser Lagrangian Hessian matrix, which led to slightly higher function evaluation times. This difference arises due to the

**Table 4. Comparison of different optimisation algorithms using IPOPT. The optimisation across platforms was performed on the same optimisation model, using the lockdown case scenario as an example ($n = 100$ runs, $dt = 0.5$). Hessian nnz denotes the number of non-zero elements of the Lagrangian Hessian matrix. Objective function evaluations (183) and Lagrangian Hessian evaluations (128) were identical across all platforms.**

| Language | Interface | Iters | Overall time (s) | Func. eval time (s) | Hessian nnz | Objective |
|----------|-----------|-------|------------------|---------------------|-------------|-----------|
| Julia | JuMP | 128 | 0.1468 ± 0.0127 | 0.0145 ± 0.0023 | 1200 | 0.6332 |
|  | AmplNLWriter | 128 | 0.1525 ± 0.0120 | 0.0097 ± 0.0008 | 1000 | 0.6332 |
| Python | Pyomo | 128 | 0.1543 ± 0.0059 | 0.0092 ± 0.0006 | 1000 | 0.6332 |
|  | amplpy | 128 | 0.1537 ± 0.0032 | 0.0090 ± 0.0002 | 1000 | 0.6332 |
| R | rAMPL | 128 | 0.1598 ± 0.0078 | 0.0092 ± 0.0006 | 1000 | 0.6332 |
| AMPL | AMPL IDE | 128 | 0.1598 ± 0.0084 | 0.0093 ± 0.0010 | 1000 | 0.6332 |

automatic differentiation approach used by each platform to compute the Hessian sparsity pattern. In JuMP, the automatic differentiation framework identified a comparatively denser sparsity structure for the Lagrangian Hessian. In contrast, the remaining algorithms depend on the AMPL Solver Library (ASL) [37] for the derivative evaluations, either directly (amplpy, rAMPL, and AMPL IDE), or indirectly (Pyomo and AmplNLWriter), which exploits partial separability to yield a more compact sparsity representation [38].

From the optimisation algorithms tested in this study, JuMP and Pyomo are open-source, while amplpy and rAMPL are official interfaces that allow the user to access the license-based optimisation language AMPL and its features from Python and R, respectively. On the other hand, AmplNLWriter is an open-source wrapper that also interacts with AMPL-enabled solvers but is maintained by the JuMP community. Importantly, AmplNLWriter does not require an AMPL license when used with open-source solvers like IPOPT; however, a license may be needed to use AMPL's proprietary solvers.

By applying the solver to the initial case scenario using multiple programming languages, we demonstrate that users can choose their preferred programming language while utilising either open-source or licensed optimisation platforms, depending on their needs.

Additionally, we compared the performance of JuMP with alternative NLP solvers, including MadNLP [39,40] and UnoSolver [41], two alternative open-source NLP solvers. Applied to Case Scenario 1 and Case Scenario 4, all solvers converged to equivalent objective values, confirming that the solutions are not solver-dependent. Full benchmark results and trajectory comparisons are available in the accompanying repository.

## Discussion

This study demonstrates that existing direct nonlinear programming methods can be effectively applied to epidemiological optimal control problems without introducing new optimisation theory. By leveraging modern optimisation tools such as JuMP, previously published compartmental epidemic models can be readily reformulated and solved using direct optimisation approaches. The results show that optimal control strategies can be computed without deriving adjoint equations or implementing custom numerical solvers. Taken together, these findings indicate that direct optimisation methods provide a practical and accessible framework for applying optimal control in epidemiological modelling, with potential relevance for decision-making and policy analysis.

Modelling underpinned a number of key policy decisions in the UK during the COVID-19 pandemic. As in the examples here, the timing, duration, and extent of lockdown were foremost among the policy options considered at several time points [42]. In trying to provide rapid real-time advice, COVID-19 modelling approaches often relied on combining scenarios based on systematic grid searches of assumed plausible parameters [43,44]. Furthermore, the formal consideration of costs to address questions of optimality has been identified as a priority for future modelling in the UK COVID-19 Inquiry.

The Core Decision-Making Report noted it was not clear to what extent 'smarter' interventions (that maximised the impact on transmission while minimising the economic or social impact) were ever seriously taken forward, thus hampering the ability of decision-makers to assess and balance relative harms [45]. With tight time constraints involved in preparing modelling output for real-time decision-making, the ability to automate the search of policy options in a manner that can both save time and identify optimal choices offers considerable advantages, especially for complex models with long runtimes that are often used in informing decision-making.

The application of numerical simulations and control theory is commonplace in the analysis of complex models, especially in fields dealing with unpredictable dynamics and large-scale systems [18]. While these simulations provide valuable insights into a wide range of complex systems, optimal control theory becomes particularly helpful when managing disease control, as it helps balance resource constraints and optimise treatment or control strategies. In addition, these strategies also aim to reduce the implementation costs associated with large-scale interventions, offering a double benefit of improving public health outcomes while minimising economic impact [1,17].

The continuous nature of many models, including those used in disease control, often necessitates discretisation for computational implementation. This study employed a simple Euler discretisation, a straightforward method for approximating continuous systems. While easy to implement, this method has limitations regarding accuracy and stability. We conducted a timestep sensitivity analysis comparing simple Euler discretisation with an exponential approximation approach (where transitions are modelled as $1 - \exp(-\lambda \cdot dt)$) across six timestep values for Case Scenario 1. Both methods converged successfully across all timesteps tested, with the simple Euler method showing greater stability (1.58% relative variation) compared to exponential discretisation (22% relative variation). At small timesteps, both methods reached nearly identical optimal solutions (0.7% difference at dt = 0.05). These results demonstrate that for the models considered here, Euler discretisation with appropriately chosen timesteps provides reliable convergence S3 Fig.

In Case Scenario 1, the lockdown intervention was optimised; it is worth noting that other grid-search-like methods could have been employed. For instance, one approach could involve fixing the intervention length to 20 days and optimising only the start of the intervention. The supplementary S1 Fig, shows a comparison of the optimal time found using JuMP with the cumulative incidence obtained by running the model simulations for different intervention start times. However, the use of JuMP allowed us to confirm the optimal intervention frequency and duration.

Compared to Case Scenario 1, where the objective was to minimise the overall number of infections, Case Scenario 2 had a different focus, the goal being to keep the infected population below a certain threshold, which could be influenced by factors such as the current healthcare capacity. This led to a distinct optimal policy, which suggested a single lockdown period that started earlier than in scenario 1. However, unlike in scenario 1, the strength of the intervention gradually decreased over time until the effective reproduction number dropped below one, at which point disease transmission was effectively controlled. While this optimal policy provides a theoretical framework for managing disease transmission, there are significant challenges in translating these findings into actual intervention policies. Fine-tuning the intensity of the intervention over time, as suggested by the model, may not be feasible in real-world settings. Instead, a more practical approach may involve implementing a series of staged interventions with varying intensities, depending on the evolving situation. The impact of the intervention may be uncertain prior to its implementation, and if its efficacy is found to be lower than expected, it may require initiating the intervention well in advance before reaching the infected threshold. Furthermore, determining when to stop the intervention requires knowledge of the effective reproduction number in the absence of the intervention. This requires reliable estimates of $R_t$ and the intensity of the intervention. These uncertainties are in addition to the usual uncertainty in model structure and parameter values of the underlying model.

Similarly, Case Scenario 3 relies on the assumption that a vaccine is available at the start of an epidemic, and in such a case, the optimal policy would be to vaccinate early and at the maximum capacity available until the supply is exhausted, to reduce the susceptible fraction of population as much as possible. However, it is important to recognise that, in reality, even if a vaccine is developed quickly, there are several factors that can delay or limit its availability. The production

capacity may be insufficient to meet demand, leading to delays in distribution. Additionally, the vaccine's efficacy may not be immediately known or consistent across all demographic groups. Variations in immune response based on age, health status, or pre-existing conditions could affect the overall success of the vaccination campaign.

The optimal control trajectories obtained in this study correspond to solutions under specific modelling assumptions. Control variables are normalised and bounded, providing a structured way to explore the relative timing and intensity of interventions, rather than literal representations of policy actions. In practice, the realised impact of an intervention depends on behavioural responses, compliance, enforcement, and contextual factors that are difficult to quantify precisely, and this uncertainty is well recognised [45]. Within this context, the value of direct optimisation lies in its ability to support structured exploration under uncertainty.

Despite the obvious appeal of identifying optimal control strategies, there are challenges associated with solving optimisation problems. We highlight here that indirect methods are more mathematically intensive, and their successful application depends on an in-depth understanding of the system's dynamics and constraints. In contrast, the direct methods described do not require the explicit construction of the adjoint and control equations, making them more straightforward to define computationally. Nevertheless, they may come with limitations, such as potentially slower convergence or less precision in highly complex systems. Another issue may arise when the constraints of an NLP problem turn out to be infeasible, which is a common problem that the user should be prepared to address. In these cases, the solver may fail to converge to a feasible solution, which may require revisiting and adjusting the model or constraints [18]. Thus, it is essential to have a clear understanding of the system and the ability to adjust the model as necessary to avoid such pitfalls. Therefore, we do not advocate one approach as the standard; rather we aim to demonstrate to the reader how optimisation languages such as JuMP can be a very useful tool to obtain fast optimal approximations or even corroborate the results obtained through indirect methods. Similarly, Silva et al. [46] applied both direct and indirect methods to develop optimal control strategies to minimise the cost of interventions for treating tuberculosis. They utilised AMPL, IPOPT, and PROPT MATLAB Optimal Control Software to compute and compare the numerical results with those obtained through an iterative method, a standard approach for solving systems of ODEs and updating controls in indirect methods.

The complexity of deriving adjoint equations and Hamiltonians increases significantly with model dimension, making indirect methods more challenging for larger models. For instance, the simple exponential infection model discussed earlier requires only a single adjoint equation and one control optimality condition. However, when we consider a 3-compartment SIR model, the situation becomes more complex, requiring three adjoint equations, three boundary conditions, and coupling between state and adjoint variables. In larger models, such as the dengue transmission model (Case Scenario 4), which includes seven compartments ($S_h$, $I_h$, $I_{hA}$, $P$, $R_h$, $S_v$, $I_v$) and four control variables ($u_1$, $u_2$, $u_3$, $u_4$), the analytical derivation becomes much more complicated. In this case, seven adjoint equations must be derived (one for each state variable), along with four control optimality conditions. Additionally, the boundary conditions become more intricate, and the coupling between state and adjoint variables increases nonlinearly as the model dimension grows. This analytical burden becomes increasingly tedious and prone to error as models become more complex, requiring meticulous manual derivation, verification, and implementation. In contrast, direct methods only require the discretisation of the state equations, the definition of variables and constraints, and then the problem can be passed to a solver. The complexity in direct methods scales approximately linearly with the number of time points and compartments. This scalability advantage becomes particularly evident for models that incorporate additional complexities such as age structure, spatial heterogeneity, or multiple pathogen strains.

As for solving NLP optimisation problems, using an interior point solver such as IPOPT has proven to be highly effective in this study and in others [19,33,46–48]. IPOPT can efficiently solve problems with a large number of inequalities and degrees of freedom, making it a good and flexible option for high-dimensional control problems. The algorithm's ability to scale and find feasible solutions within a reasonable time frame is an important advantage when tackling the optimisation challenges posed by complex control systems. Additionally, advancements in computing power have reduced the

computational expense once associated with these methods, enabling faster and more accessible solutions. However, alternative tools such as AMPL, GAMS, and Pyomo also offer similar capabilities, providing users with flexibility in software selection based on preference and the specific requirements of their computational environment.

All presented case studies, including the most complex example (Case Scenario 4: a 7-compartment dengue model with four controls, which, after discretisation, resulted in 5,409 decision variables and 4,215 constraints), converged successfully using default IPOPT settings within reasonable computation times. However, we did not systematically evaluate sensitivity with respect to initialisation strategies, problem scaling, stiffness, solver tolerances, or parameter uncertainty. While JuMP provides a high-level interface for formulating nonlinear optimisation problems, convergence behaviour is ultimately determined by the underlying solver and the numerical properties of the discretised model [31]. Interior-point methods are known to exhibit sensitivity to numerical choices such as starting points and scaling, particularly in more challenging settings [49]. The models considered here are well-behaved. In contrast, more complex formulations incorporating spatial structure, stochastic effects, population heterogeneity, or multiple strains may introduce additional numerical difficulties. In such cases, the direct optimisation framework remains conceptually applicable. Still, increased model complexity primarily raises computational considerations, including larger numbers of decision variables and constraints, potential numerical stiffness, and the need for solver tuning or alternative discretisation strategies. Addressing these challenges may require additional numerical measures, such as custom scaling, higher-order discretisation schemes (i.e., Runge-Kutta [50]), or multiple initialisation strategies to improve robustness and convergence [18,19]. It is worth noting that, while we employed Euler discretisation throughout this study, the framework itself does not specify a specific discretisation approach; modellers can implement alternative schemes as needed, provided the resulting formulation remains differentiable and compatible with automatic differentiation.

While this work focuses on direct nonlinear programming (NLP) transcription methods for optimal control, other approaches can provide complementary capabilities. Model Predictive Control (MPC) formulates a sequence of finite-horizon optimisation problems in a receding-horizon framework, making it well-suited for multivariable dynamic systems with constraints and for adapting to real-time disturbances [51,52]. In contrast, direct NLP transcription discretises the full-time horizon into finite elements and formulates a single finite-dimensional optimisation problem, in which the system dynamics and control variables are solved simultaneously [18,53]. This approach offers advantages in exploiting sparsity and handling complex path and boundary constraints. However, for nonconvex problems, the resulting nonlinear programs are typically solved using optimisation solvers such as IPOPT, which provide local rather than theoretical global optimality [19].

Reinforcement learning (RL) and deep learning–based control methods learn control policies through trial-and-error interaction with an environment and can be effective for systems with stochastic behaviour or large state spaces [54–57]. In these settings, traditional control approaches attempt to precompute control actions for all possible system states, which often become impractical as the problem size increases. This difficulty arises because the complexity of such controllers grows rapidly with the number of system variables and constraints. RL addresses this challenge by using flexible function approximators to represent control policies, allowing it to scale to higher-dimensional problems. However, this flexibility typically requires large amounts of training data and comes with reduced interpretability and weaker safety or stability guarantees compared to model-based methods. In comparison, the direct NLP framework provides a transparent middle ground between classical analytical methods and data-driven approaches, making it well-suited for deterministic models with well-characterised system parameters.

Additionally, this study focused exclusively on differentiable epidemiological models, which allowed us to use gradient-based optimisation solvers. As a result, our findings do not directly address the effectiveness or applicability of non-gradient-based optimisation approaches. However, recent advancements have introduced differentiable frameworks for traditionally non-differentiable models, such as agent-based models (ABMs), which may expand the scope and use of gradient-based optimisation solvers. For instance, Chopra et al. [58], developed GradABM, a scalable and differentiable design for ABMs that enables gradient-based learning through automatic differentiation.

## Conclusion

Overall, the NLP optimisation approach illustrated in this study performed very effectively for simple models, consistently delivering accurate results within seconds which showcase its practical value as an accessible tool for rapid analysis and decision-making in epidemiological modelling.

While JuMP and IPOPT have been used in previous studies to solve optimisation problems in the context of infectious disease modelling [48], this study aims to showcase their ease of use and effectiveness to promote the wider adoption of direct methods for addressing optimisation challenges. By employing JuMP and IPOPT, this work illustrates how these tools can effectively deliver an initial solution. This approach can aid policymakers in making informed decisions during the early stages of an epidemic, based on the dynamics of the disease model.

Real-world policy questions typically concern trade-offs between timing, duration, and resource constraints, rather than exact intervention intensities. Direct optimisation methods allow these constraints to be represented explicitly and modified as assumptions evolve. When implemented using frameworks such as JuMP, models can be reformulated and re-solved efficiently to examine how conclusions change under different operational constraints or effectiveness assumptions. This flexibility enables direct optimisation to function as a practical decision-support tool, facilitating comparative and iterative analysis rather than producing fixed solutions.

Regarding ongoing challenges in public health economics, the uncertainty surrounding costs has been consistently highlighted in the evidence presented to the UK COVID Inquiry [45]. As we consider future developments, it is important to integrate cost considerations into our decision-making processes. Despite this need, it is likely that cost estimates will continue to be uncertain. In addition, there may be other considerations that would affect the optimal policy; for example, Zarebski et al. [59] incorporated measures of equity into finding an optimal vaccination program in order to achieve an intervention that was both effective and ethical. Thus, there is a pressing need for efficient methodologies that facilitate the exploration of optimal strategies across a wide array of cost function assumptions. The methods described here offer an opportunity for real-time evaluation and the potential generation of rules of thumb for optimal approaches, based on extensive exploration of cost-benefit analysis under a wide range of disease dynamic scenarios, to support preparedness.

## Supporting information

**S1 Fig. Impact of intervention timing on fixed lockdown duration.** Final cumulative infected population values (C) obtained from applying a fixed intervention with a length of 20 days at different starting times. The minimal C value is highlighted in orange.
(TIF)

**S2 Fig. Optimal intervention strategy across multiple parameter scenarios.** Performance of an optimal intervention strategy to reduce the cummulative infections, as in Case Scenario 1, across various scenarios by sweeping through a range of transmission and recovery parameter values ([0.35,0.55] for $\beta$ and [0.15,0.30] for $\gamma$). Rather than optimising for each scenario separately, the JuMP model creates a unified intervention profile that reduces the expected total infections across all.
(TIF)

**S3 Fig. Timestep sensitivity analysis.** Comparison of simple Euler and exponential discretisation methods across six timestep values (dt = 2.0 to 0.05 days) for Case Scenario 1. (a) Shows the convergence of the optimal objective value with timestep for both methods. (b) Displays the relative difference between methods. (c) Illustrates the scaling of computational time with the number of time points (mean ± standard deviation across 50 repetitions). (d) Compares control trajectories at dt = 0.5 days.
(TIF)

## Acknowledgments

The authors sincerely thank Dr. Joshua Asamoah, whose shared files and communication supported the testing of the methods presented in this study.

## Author contributions

**Conceptualization:** Sandra Montes-Olivas, Simon D.W. Frost.

**Formal analysis:** Sandra Montes-Olivas.

**Funding acquisition:** Michael B. Gravenor, Simon D.W. Frost.

**Investigation:** Sandra Montes-Olivas.

**Methodology:** Sandra Montes-Olivas, Simon D.W. Frost.

**Software:** Sandra Montes-Olivas.

**Supervision:** Simon D.W. Frost.

**Visualization:** Sandra Montes-Olivas.

**Writing – original draft:** Sandra Montes-Olivas, Simon D.W. Frost.

**Writing – review & editing:** Adam J. Kucharski, Michael B. Gravenor, Simon D.W. Frost.

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
