## [Decision Letter · Decision Letter 0]

5 Dec 2025

PCOMPBIOL-D-25-02022

Exploring epidemic control policies using nonlinear programming and mathematical models

PLOS Computational Biology

Dear Dr. Montes-Olivas,

Thank you for submitting your manuscript to PLOS Computational Biology. After careful consideration, we feel that it has merit but does not fully meet PLOS Computational Biology's publication criteria as it currently stands. Therefore, we invite you to submit a revised version of the manuscript that addresses the points raised during the review process.

We look forward to receiving your revised manuscript.

Kind regards,

Joseph T. Wu

Academic Editor

PLOS Computational Biology

Denise Kühnert

Section Editor

PLOS Computational Biology

**Journal Requirements:**

3) We notice that your supplementary Figures are included in the manuscript file. Please remove them and upload them with the file type 'Supporting Information'. Please ensure that each Supporting Information file has a legend listed in the manuscript after the references list.

4) Please amend your detailed Financial Disclosure statement. This is published with the article. It must therefore be completed in full sentences and contain the exact wording you wish to be published.

**Reviewers' comments:**

Reviewer's Responses to Questions

**Comments to the Authors:**

Reviewer #1: The topic is potentially valuable, particularly given the increased interest in optimization for epidemic response during the COVID-19 pandemic. The manuscript is well written, and the examples are clear and reproducible. The current version reads more like tutorial or software demonstration rather than a research article. The examples rely on low-dimensional “toy” systems, and important issues such as discretization robustness, scalability, and real-world policy constraints need further discussion. The use of extremely simplified models, the absence of rigorous numerical analysis, and the lack of engagement with the real complexities of epidemic control significantly limit the impact. In its present form, the novelty is modest and several essential issues remain insufficiently addressed.

Reviewer #2: Review is uploaded as an attachment

**Have the authors made all data and (if applicable) computational code underlying the findings in their manuscript fully available?**

Reviewer #1: Yes

Reviewer #2: Yes

PLOS authors have the option to publish the peer review history of their article (what does this mean?). If published, this will include your full peer review and any attached files.

Reviewer #1: **Yes:** Mohamed A. Bakheet

Reviewer #2: No

**Figure resubmission:**
---

## [Decision Letter · Decision Letter 1]

14 Apr 2026

Dear Dr Montes-Olivas,

We are pleased to inform you that your manuscript 'Exploring epidemic control policies using nonlinear programming and mathematical models' has been provisionally accepted for publication in PLOS Computational Biology.

Best regards,

Joseph T. Wu

Academic Editor

PLOS Computational Biology

Denise Kühnert

Section Editor

PLOS Computational Biology

Reviewer's Responses to Questions

**Comments to the Authors:**

Reviewer #1: Thank you for the careful revision and thorough point-by-point response. The manuscript has improved substantially since the previous version, and I appreciate the effort the authors have made to address the major concerns. The paper is now framed much more clearly as an accessibility- and implementation-focused contribution rather than as a fundamentally new optimization-methods paper, which makes the contribution more accurate, balanced, and easier to evaluate on its own terms. I also found the expanded discussion of practical policy constraints, solver settings, convergence behavior, and alternative approaches such as model predictive control and reinforcement learning to be helpful. In particular, the timestep sensitivity analysis strengthens the numerical section and provides useful reassurance regarding the use of Euler discretization.

My only remaining reservation is that scalability is still supported more by discussion and motivation than by direct demonstration. The manuscript continues to rely on relatively simple illustrative models, and while the added explanation about possible extension to higher-dimensional, spatial, or stochastic systems is reasonable and valuable, it does not yet fully show how the framework would perform in those more complex settings. That said, I do not view this as a major weakness in the current revision. Rather, it suggests that the manuscript is best understood as a practical and pedagogical methods contribution with clear applied value, while broader claims about scalability could be explored further in future work. Overall, the revision is thoughtful and responsive, the claims are now much better calibrated, and my main concerns have largely been addressed.

Reviewer #2: I'd like to thank the authors for their detailed responses to my concerns. Overall, I think the manuscript is much improved and I am satisfied with how they addressed my major concerns and the current state of the manuscript. I have a few minor comments:

Line 160-161: it would be helpful to have a citation to support the statement that JuMP is "efficient for large-scale optimization problems"

Line 276: there are a few typos concerning subscripts

Lines 285-286: I think it would be more appropriate to say that, from an epidemiological standpoint, the primary goal is to reduce the number of susceptible individuals below a level that allows for maintaining transmission.

Lines 306-307: This sentence reads a bit oddly to me, you may want to consider rewording it.

Line 309 and Equations 10 and 11: Did you intend for the A in the asymptomatic state variable to be subscript?

**Have the authors made all data and (if applicable) computational code underlying the findings in their manuscript fully available?**

Reviewer #1: Yes

Reviewer #2: None

PLOS authors have the option to publish the peer review history of their article (what does this mean?). If published, this will include your full peer review and any attached files.

Reviewer #1: **Yes:** Mohamed Bakheet

Reviewer #2: No

---

## [Editor Report · Acceptance letter]

PCOMPBIOL-D-25-02022R1

Exploring epidemic control policies using nonlinear programming and mathematical models

Dear Dr Montes-Olivas,

I am pleased to inform you that your manuscript has been formally accepted for publication in PLOS Computational Biology. Your manuscript is now with our production department and you will be notified of the publication date in due course.

With kind regards,

Zsofia Freund
